# OMNIAGENT: LONG-VIDEO GENERATION VIA CROSS-MODAL MULTI-AGENT ORCHESTRATION

## ABSTRACT

Recent advancements in multi-agent systems have demonstrated significant potential for enhancing creative task performance, such as long video generation. This study introduces three innovations to improve multi-agent collaboration. First, we propose OmniAgent, a hierarchical, graph-based multi-agent framework for long video generation that leverages a film-production-inspired architecture to enable modular specialization and scalable inter-agent collaboration. Second, inspired by context engineering, we propose hypergraph nodes that enable temporary group discussions among agents lacking sufficient context, reducing individual memory requirements while ensuring adequate contextual information. Third, we transition from directed acyclic graphs (DAGs) to directed cyclic graphs with limited retries, allowing agents to reflect and refine outputs iteratively, thereby improving earlier stages through feedback from subsequent nodes. These contributions lay the groundwork for developing more robust multi-agent systems in creative tasks.

## 1 INTRODUCTION

Generating minute-scale, coherent videos from text that satisfy user intent—while maintaining precise control over intermediate scripts, keyframes, audio, and the final cut—requires multiple specialized agents to effectively cooperate. Success hinges less on any single backbone and more on orchestration: how role-specific agents coordinate, what context they can access, and whether feedback can propagate backward to refine upstream decisions. Flat dispatch patterns (e.g., a single Director calling specialists) and strictly acyclic pipelines struggle to model cross-stage dependencies, to share the right context at the right time, and to support reflection once downstream modules surface inconsistencies. These gaps are acute in creative pipelines where script, storyboard, cinematography, sound, and editing must evolve together under changing constraints.

We introduce **OmniAgent**, a cross-modal multi-agent framework that elevates the orchestration layer. **OmniAgent** organizes agents in a hierarchical graph, equips them with a context-engineering mechanism based on transient hypergraph collaboration, and allows bounded cyclic execution for graph-level reflection under a retry budget. The framework couples LLM planning with image/video/audio back-ends and a video-understanding module, mirroring real filmmaking workflows while remaining computationally disciplined.

Our work draws motivation from both the collaborative workflows of real-world creative teams and the practical demands of context engineering in LLM-based multi-agent systems (MAS):

**(1) Hypergraph-based context collaboration.** In real-world creative projects, when an individual lacks critical context—such as visual style or narrative intent—they don't work in isolation; instead, they call a quick team huddle to gather insights. Inspired by this, OmniAgent allows any agent to dynamically convene a temporary "team meeting" with relevant peers whenever its current context is insufficient. This collaborative retrieval enriches decision-making without overloading any single agent's memory, effectively distributing knowledge across the system just like human teams do.

**(2) Bounded cycles for reflection.** Real creative workflows are rarely linear—feedback, revisions, and even rework are common as teams refine their output. Drawing from this iterative nature, OmniAgent moves beyond rigid, acyclic execution by allowing limited feedback from downstream to upstream agents (e.g., a video generator flagging a continuity issue back to the scriptwriter). To avoid endless loops, each feedback edge is governed by a small retry budget, enabling up to a few

rounds of productive reflection—mirroring how human teams iterate toward higher quality without getting stuck.

**(3) OmniAgent.** Traditional multi-agent systems for content creation typically adopt one of two oversimplified approaches: either a linear sequence of tasks or a centralized "director" agent that manages all operations. Neither approach adequately reflects collaborative nature of real-world filmmaking. OmniAgent overcomes this limitation by organizing agents into a hierarchical structure that closely mirrors established film production pipelines—spanning concept development, scriptwriting, storyboarding, asset generation, script supervising and post-production.

We evaluate five video-generation versions—two commercial baselines and three of our own (flat, hierarchical w/o context engineering, and full)—on three single-sentence text prompts (average 36 words; target duration ∼1 minute). Human quality is judged with the *FilmEval* rubric (six dimensions; 12 items) by 12 audience raters and 4 experts in a within-subject design with counterbalancing. Empirically, hierarchy preferentially improves structure-centric dimensions—Narrative & Script (NS) and Rhythm & Flow (RF)—while adding context engineering + bounded cycles further lifts Audiovisuals/Techniques (AT), Aesthetics/Expression (AE), and Engagement (EE), with the full system attaining the best pooled Overall Experience (OE).

**Contributions.**

- We propose **OmniAgent**, a novel hierarchical, graph-based multi-agent framework for long video generation, where agents possess private memory and engage in structured, dynamic communication. In contrast to prior approaches that rely on either simplistic agentic workflows (e.g., sequential task chaining) or highly centralized multi-agent architectures (e.g., a single director agent coordinating all others), OmniAgent adopts a *hierarchical organization* inspired by real-world film production pipelines, enabling modular specialization, inter-stage coordination, and scalable collaboration.

- We introduce a **hypergraph-based context retrieval** mechanism that enables on-demand, collaborative knowledge gathering across agents, balancing context richness with memory efficiency.

- We design a **controlled cyclic execution** strategy with retry budgets, allowing limited backward edges for iterative refinement and reflection while preventing infinite loops—enabling failure-aware, multi-round video production.

## 2 OMNIAGENT

In this section, we present **OmniAgent**, a novel multi-agent framework designed for long video generation. **OmniAgent** introduces a hierarchical, graph-based architecture where each agent operates with its own memory and can dynamically interact with others through structured communication. The framework integrates large language models (LLMs), multimodal foundation models, and a novel context engineering strategy to enable complex, iterative, and collaborative video production workflows that mirror real-world filmmaking pipelines.

### 2.1 AGENT AND MULTI-AGENT GRAPH FORMALISM

We formalize **OmniAgent** as a **directed graph** $G = (V, E)$, where each node $v_i \in V$ represents an autonomous **agent**, and each directed edge $e_{ij} \in E$ denotes a unidirectional information flow from agent $v_i$ (the *source*) to agent $v_j$ (the *target*). Unlike prior works that enforce a strict **Directed Acyclic Graph (DAG)** structureQian et al. (2024), **OmniAgent** permits limited cycles to enable reflection and iterative refinement (see Section 2.3).

Each agent $v_i$ is defined as a tuple:
$$v_i = (M_i, A_i, f_i, T_i), \tag{1}$$
where:

- $M_i$ is the **private memory** of agent $v_i$, storing its interaction history with other agents;
- $A_i \subseteq V \setminus \{v_i\}$ is the set of agents that $v_i$ has communicated with (i.e., its adjacency in the interaction subgraph);

- $f_i : C_i \to (O_i, I_{i \to *})$ is the agent's reasoning function, mapping its input context $C_i$ to an **artifact** $O_i$ (e.g., script, storyboard, video clip) and a set of **instructions** $I_{i \to *}$ for downstream agents;

- $T_i$ is the set of **tools** accessible to $v_i$ (e.g., image/video/audio generation or VLM for video understanding).

The **context** $C_i$ available to agent $v_i$ at inference time consists of two components:

$$C_i = \underbrace{\bigcup_{v_j \in A_i} \text{Dialog}(v_j \to v_i)}_{\text{Conversational Memory}} \cup \underbrace{\bigcup_{k \in \text{Pred}(i)} O_k}_{\text{Artifact Context}} , \tag{2}$$

where $\text{Pred}(i) = \{k \mid e_{ki} \in E\}$ denotes the set of immediate predecessor agents, and $\text{Dialog}(v_j \to v_i)$ records the message history from $v_j$ to $v_i$. Artifacts are consumed strictly in accordance with instructions issued by predecessor agents, ensuring a structured workflow progression.

## 2.2 HIERARCHICAL AGENT ORGANIZATION

Inspired by real-world film production pipelines, **OmniAgent** organizes agents into a **hierarchical workflow graph** that mirrors stages such as *concept development*, *scriptwriting*, *storyboarding*, *visual asset generation*, *video composition*, and *post-production*.

This contrasts with flat architectures, such as a single director agent dispatching tasks to specialized agents Zhang et al. (2025a) or agentic workflows Li et al. (2024); Wu et al. (2025), which struggle to model inter-stage dependencies and support iterative refinement.

## 2.3 CONTEXT ENGINEERING VIA HYPERGRAPH COLLABORATION AND CONTROLLED CYCLES

To address the tension between context richness and memory efficiency, we introduce two key context engineering mechanisms.

### 2.3.1 HYPERGRAPH-BASED CONTEXT RETRIEVAL

When an agent $v_i$ determines during reasoning that its current context $C_i$ is insufficient (e.g., missing visual style references or narrative constraints), it dynamically forms a **hypergraph node** $h$ comprising itself and a set of context-relevant agents $S_i \subseteq V \setminus \{v_i\}$. This simulates a "team meeting" where multiple agents collaboratively resolve the information gap.

The selection of $S_i$ follows a **recursive breadth-first search** over the agent graph:

$$S_i^{(0)} = \text{Pred}(i), \tag{3}$$

$$S_i^{(d)} = \bigcup_{v_j \in S_i^{(d-1)}} (\text{Pred}(j) \cup \text{ActiveSucc}(j)) \setminus \bigcup_{k=0}^{d-1} S_i^{(k)}, \tag{4}$$

where $\text{ActiveSucc}(j) = \{v_k \mid e_{jk} \in E \text{ and } v_k \text{ has been activated}\}$. The search proceeds to depth $d = 0, 1, 2, \dots$ until sufficient context is gathered or a maximum depth $D_{\max}$ is reached. The final collaborating set is $S_i = \bigcup_{d=0}^{D} S_i^{(d)}$, and all agents in $\{v_i\} \cup S_i$ engage in a multi-turn discussion to enrich $C_i$.

This mechanism **distributes memory** across agents, reducing per-agent context load while ensuring on-demand access to global knowledge.

### 2.3.2 CONTROLLED CYCLIC EXECUTION FOR REFLECTION

Unlike conventional DAG-based agent systems, **OmniAgent** allows **limited cyclic dependencies** to enable backward refinement. For instance, if a *Script Supervisor Agent* discovers that a generated shot violates continuity established in the script, it can trigger a revision request to the *Scriptwriter Agent*.

We model this by permitting edges that form cycles, but enforce a **retry budget** $R_{\max} = 3$ **only on reverse edges**. Formally, given an initial DAG topology that defines a partial order over agents, an edge $e_{ij}$ (from agent $i$ to agent $j$) is classified as a *reverse edge* if agent $j$ precedes agent $i$ in this topological order (i.e., information flows backward relative to the original execution direction). Only such reverse edges are subject to the retry budget.

Specifically, each reverse edge $e_{ij}$ maintains a counter $c_{ij}$ tracking the number of times information has flowed along it. If $c_{ij} \geq R_{\max}$, the edge is **temporarily disabled**, effectively converting the graph back into a DAG for that execution path. This prevents infinite loops while allowing up to three rounds of reflection. Forward edges (those consistent with the original topological order) remain unrestricted and may be traversed freely across rounds.

Formally, the execution proceeds in **rounds** $t = 1, 2, \ldots$. At each round, the active agent set $V_{\text{active}}^{(t)}$ is determined by readiness (all predecessors satisfied or retry allowed). After processing, counters for reverse edges are updated:

$$c_{ij}^{(t+1)} = \begin{cases} c_{ij}^{(t)} + 1 & \text{if } e_{ij} \text{ is a reverse edge and was traversed at round } t \\ c_{ij}^{(t)} & \text{otherwise} \end{cases} \tag{5}$$

and any reverse edge with $c_{ij}^{(t+1)} > R_{\max}$ is pruned from $E$ for subsequent rounds.

This design enables **failure-aware iteration**: agents retain memory of prior attempts (via $M_i$), allowing them to avoid repeating mistakes during retries.

### 2.4 MODELS USED

The **OmniAgent** framework is built upon the following existing models:

- **Language model**: GPT-4o is used to power agent reasoning and inter-agent communication.
- **Image generation**: Seedream 3.0Gao et al. (2025a) is used to generate static visual assets, such as character designs, environments, and storyboards.
- **Video generation**: Seedance 1.0Gao et al. (2025b) is used to generate video sequences.
- **Audio generation**: HunyuanVideo-FoleyShan et al. (2025) is used to generate sound effects and ambient audio.
- **Video understanding**: Qwen-VL-Max is used for analyzing video content.

Each agent is granted access to a subset of these models based on its designated function.

### 2.5 EVALUATION METRICS

**Instrument.** We evaluate cinematic quality using the *FilmEval* rubric, which is organized into six dimensions with twelve criteria: Narrative & Script (NS), Audiovisuals & Techniques (AT), Aesthetics & Expression (AE), Rhythm & Flow (RF), Emotional & Engagement (EE), and Overall Experience (OE); decomposed into {SF, NC, VQ, CC, PLC, V/AQ, CT, AVR, NP, VAC, CD, OQ}. We use the identical five-point Likert anchors and original item wording for both cohorts (audience and experts), as reproduced in Appendix A.1.

**Prompts and rating targets.** Each rated video is generated from one of the **three text prompts** (P1–P3) that specify content, mood, and stylistic elements for an approximately one-minute video. Raters judge the resulting video solely based on this questionnaire.

**Scoring procedure.** For each video, raters provide twelve item scores on a 1–5 Likert scale (higher is better). We then compute dimension scores by averaging their constituent items (unweighted arithmetic means):

$$\begin{aligned} \text{NS} &= \text{mean}(\text{SF}, \text{NC}), & \text{AT} &= \text{mean}(\text{VQ}, \text{CC}, \text{PLC}, \text{V/AQ}), \\ \text{AE} &= \text{mean}(\text{CT}, \text{AVR}), & \text{RF} &= \text{mean}(\text{NP}, \text{VAC}), \\ \text{EE} &= \text{CD}, & \text{OE} &= \text{OQ}. \end{aligned}$$

Unless otherwise noted, we report both *item-level* scores (12 criteria) and *dimension-level* means (6 composites).

**Aggregation for reporting.** To summarize performance per model while respecting within-subject comparisons across the three prompts, we first average, for each rater, the scores of the same model across P1–P3 (subject-level prompt average). We then aggregate these subject-level values across raters within each cohort (audience vs. experts); pooled analyses combine both cohorts.

**Transparency measures (optional).** Following common practice, we compute agreement diagnostics (e.g., audience–expert agreement, inter-rater reliability) and provide the full questionnaire, anchors, and scoring templates in the appendix. These diagnostics do not alter the definition of our primary outcomes (12 items and 6 dimension means).

## 3 EXPERIMENTS

**Prompts & task.** We evaluate **OmniAgent** on one–sentence prompts for minute-scale long-video creation, executing the full pipeline from *script* to *storyboard* to *auto-edit* and *auto-publish*. We design 3 text prompts (Prompts 1/2/3) with distinct visual and narrative styles (avg. length $\approx 36$ words), specifying content, mood, and stylistic elements for $\sim$ 1-minute videos.

**Conditions (five versions).** We compare five video-generation versions: *(1) setting1_flat* — Director–Agent flat scheduling (no hierarchy); *(2) setting2_hier_no_ctx* — hierarchical orchestration *without* context engineering; *(3) setting3_full* — our full framework with hierarchical orchestration, hypergraph-based context collaboration, and bounded cycles (retry budget) for graph-level reflection; *(4) AiPai*[1] and *(5) Video Ocean*[2] — black-box commercial baselines that support long-video generation from a single text prompt. For our three internal versions (1–3), all non-orchestration factors are held fixed (same back-ends, tool adapters, decoding temperatures, prompts, and seeds); the two commercial systems (4–5) are evaluated as-is as black-box baselines.

**Evaluation protocol.** We adopt an established short-film rubric with six dimensions and twelve criteria: *NS* (SF, NC), *AT* (VQ, CC, PLC, V/AQ), *AE* (CT, AVR), *RF* (NP, VAC), *EE* (CD), and *OE* (OQ), using identical 5-point anchors for both cohorts; see Appendix A.1 for wording and anchors. This is the same *FilmEval* instrument reproduced in our draft's Appendix A.1. Our primary outcomes are the 12 item scores and the six dimension means.

**Participants and procedure.** We recruited $N{=}16$ participants (12 audience, 4 experts) for a within-subject evaluation of five versions across three prompts. **Audience** (social media & university lists): gender $f/m{=}3/9$ (25%/75%); age distribution 18–20: $n{=}5$ (41.7%), 21–30: $n{=}5$ (41.7%), 31–40: $n{=}1$ (8.3%), 41–50: $n{=}1$ (8.3%); age $M(SD) \approx 25.3\,(8.0)$ *(estimated from bin midpoints)*; films watched (lifetime, self-reported): 0–10: $n{=}1$ (8.3%), 10–30: $n{=}2$ (16.7%), 31–60: $n{=}2$ (16.7%), 61–100: $n{=}2$ (16.7%), $> 100$: $n{=}5$ (41.7%). **Experts** (film production/studies): gender $f/m{=}2/2$; age bands: $n{=}2$ aged 31–40, $n{=}2$ aged 21–30 (exact ages not collected); professional experience (years) $M(SD){=}8.5\,(3.0)$; all $\geq 4$ years (individual: 10, 10, 10, 4).

Each participant evaluated the three prompts (1/2/3). For each prompt, they watched five videos (two commercial baselines + three of ours), for $3{\times}5{=}15$ videos per participant. Prompt order was counterbalanced with a $3{\times}3$ Latin square; within each prompt, the five-version order followed a Williams design ($5{\times}5$ Latin square) to balance first-order carryover and position effects. Prompt–version pairings were rotated so that no prompt systematically co-occurred with any version position. All videos were scored using the same questionnaire (Appendix A.1); we report cohort-specific (audience vs. experts) and pooled analyses.

**Reporting.** For internal versions (1–3), results are averaged over prompts and seeds with identical generation back-ends and hyperparameters; commercial baselines (4–5) are reported as black-box references. We report human ratings only, and additionally provide audience–expert agreement and inter-rater reliability (e.g., Cohen's $\kappa$) in Appendix A.1.

---

[1] https://aipai.ai/
[2] https://video-ocean.com/en

Table 1: Pooled across prompts P1/P2/P3: mean scores by model and dimension (higher is better).

| Model | NS | AT | AE | RF | EE | OE |
|---|---|---|---|---|---|---|
| aipai | 2.72 | 2.72 | 2.44 | 2.83 | 2.39 | 2.47 |
| video_ocean | 2.62 | 2.66 | 2.33 | 2.61 | 2.33 | 2.28 |
| setting1_flat | 2.96 | 3.00 | 2.62 | 2.93 | 2.58 | 2.75 |
| setting2_hier_no_ctx | **3.08** | 2.90 | 2.69 | **3.07** | 2.75 | 2.78 |
| setting3_full | 2.96 | **3.01** | **2.93** | 2.97 | **3.06** | **2.86** |

Bold indicates the column-wise maximum per dimension.

## 4 RESULTS

### 4.1 AUDIENCE EVALUATION (PROMPTS P1–P3)

We recruited a de-duplicated audience cohort ($n$=12) and evaluated five models within-subject under each prompt: two commercial baselines *aipai* and *video_ocean*, and our three variants, *setting1_flat* (Director-Agent flat scheduling), *setting2_hier_no_ctx* (hierarchical design w/o context engineering), and *setting3_full* (full framework). Each participant rated one video per model for prompts P1–P3 (all prompts complete: $n_{P1}$=$n_{P2}$=$n_{P3}$=12). Ratings followed the *FilmEval* instrument with six dimensions and twelve items (NS/AT/AE/RF/EE/OE; see Appendix A.1).[3] For each prompt, we ran within-subject *Friedman* tests across models per dimension, and the paired *Wilcoxon* (Holm) method to compare pooled commercial baselines (mean of *aipai*, *video_ocean*) against our approaches, pooled (mean of *setting1/2/3*) per dimension. We pooled P1-P3 by first averaging each rater's scores across prompts per model (subject-level prompt average) and then summarizing the model means per dimension.

**Results.** (*i*) *Within-prompt model effects* (Table 5): P3 shows a significant model effect on AT ($\chi^2$=9.60, $p$=0.048) and trend-level effects on AE/EE; P1 shows a trend-level effect on RF.
(*ii*) *Commercial baselines vs. Ours* (Table 6): In P2, Ours > Baselines on RF/EE/OE ($p$≤.037); in P3, Ours > Baselines on NS/AE/OE ($p$≤.045), with trend-level gains on AT/RF/EE; in P1, Baselines > Ours on RF ($p$=0.030).
(*iii*) *Pooled across prompts* (Table 1): *setting3_full* attains the highest pooled means on AT/AE/EE/OE, while *setting2_hier_no_ctx* leads NS/RF.

### 4.2 EXPERT EVALUATION

Four expert raters evaluated five models within-subjects per prompt: two commercial baselines *aipai* and *video_ocean*, and our three variants *setting1_flat* (Director-Agent flat scheduling), *setting2_hier_no_ctx* (hierarchical design without context engineering), and *setting3_full* (our full framework). Following our film evaluation protocol, we aggregated 12 items into six dimensions: **NS** (Script Faithfulness, Narrative Coherence), **AT** (Visual Quality, Character Consistency, Physical Law Compliance, Voice/Audio Quality), **AE** (Cinematic Techniques, Audio–Visual Richness), **RF** (Narrative Pacing, Video–Audio Coordination), **EE** (Compelling Degree), and **OE** (Overall Quality).

**Analysis.** Because $n$=4 is small and item distributions can deviate from normality, we used the Friedman test (nonparametric repeated-measures) per prompt and dimension (5 model levels). We report the chi-square statistic ($\chi^2$), degrees of freedom ($df$=4), $p$-value, and Kendall's W ($\eta_c$) as effect size, where $\eta_c$=$\chi^2/[N \cdot (k-1)]$ with $N$ raters and $k$=5 models. When informative, we complemented omnibus tests with pairwise Wilcoxon signed-rank tests (Holm correction). We also compared the pooled commercial baselines (*aipai*+*video_ocean*) to the pooled ours (*setting1/2/3*) using paired Wilcoxon tests per dimension and prompt. Significance thresholds: $^*p$<.05, $^\dagger p$<.10.

**Results.** (*i*) *Within-prompt omnibus tests* (Table 7): Prompt A shows a near-significant difference for **NS** ($\chi^2$=9.389, $p$=0.052, $\eta_c$=0.587), with other dimensions non-significant. Prompt B shows no

---

[3]Prompts are single-sentence textual descriptions (avg. 36 words) specifying content, mood, and stylistic elements for ∼1-minute videos; they probe the ability to translate high-level text into coherent, stylistically consistent long videos.

significant differences (largest trend on **OE**: $p=0.098$). Prompt C exhibits robust differences on **EE** ($\chi^2=13.723$, $p=0.008$, $\eta_c=0.858$), **NS** ($\chi^2=12.121$, $p=0.016$, $\eta_c=0.758$), and **OE** ($\chi^2=11.015$, $p=0.026$, $\eta_c=0.688$), with **AE** at trend level ($p=0.052$). Descriptively, medians indicate that *setting3_full* leads NS/AE in Prompt C, while *setting2_hier_no_ctx* joins *setting3_full* on RF. (*ii*) *Baselines vs. ours* (Table 8): In Prompts A/B no significant differences emerge (all $p\geq.125$). In Prompt C, the pooled ours exceed baselines by large mean margins on NS/AE/EE/OE (e.g., $\Delta M_{\text{EE}}=+1.50$), but Wilcoxon tests remain non-significant at $n=4$ (all $p\approx.125$). (*iii*) *Pooled across Prompts 1/2/3* (Table 2): Treating each *expert×prompt* as a block ($N=12$), **AE** is significant across models ($\chi^2=9.622$, $p=0.047$, $\eta_c=0.200$), suggesting robust, prompt-general advantages of our design on camera/AV expressivity.

Table 2: Experts: pooled across 1/2/3 using *expert×task* blocks ($N=12$). Friedman tests per dimension (5 models).

| Dimension | $\chi^2$ | df | p | $\eta_c$ |
|---|---|---|---|---|
| AE | 9.622 | 4 | $0.047^*$ | 0.200 |
| NS | 7.266 | 4 | 0.122 | 0.151 |
| EE | 6.412 | 4 | 0.170 | 0.134 |
| RF | 6.124 | 4 | 0.190 | 0.128 |
| OE | 5.556 | 4 | 0.235 | 0.116 |
| AT | 4.973 | 4 | 0.290 | 0.104 |

**Takeaways.** Expert judgments align with audience trends but are more conservative under $n=4$. Task C shows clear expert preference for our methods—especially *setting3_full*—on engagement (EE) and overall quality (OE), with strong narrative advantages (NS). Pooled analyses further indicate a robust, task-general benefit on camera/AV expressivity (AE). Given small-$n$ nonparametric tests (ties, zero-differences), we therefore emphasize median profiles and effect size ($\eta_c$) alongside $p$-values; full pairwise Wilcoxon (Holm-corrected) tables are included in the supplement.

### 4.3 ABLATION STUDY

**Design.** We isolate the contribution of orchestration by comparing three internal versions under identical back-ends and decoding settings: *setting1_flat* (Director-Agent flat dispatch), *setting2_hier_no_ctx* (hierarchical orchestration without context engineering), and *setting3_full* (hierarchical orchestration with hypergraph-based context collaboration and bounded cyclic reflection). Human evaluation follows the *FilmEval* rubric (six dimensions; twelve items) across three single-sentence text prompts (P1–P3). *All instructions, anchors, and scoring protocols are identical across cohorts (audience and experts).*

**Protocol.** For each rater, we first average the scores of the same model across prompts (subject-level prompt average), then summarize across raters. To combine cohorts, we pool audience ($n=12$) and expert ($n=4$) ratings via group-size weighting and pooled variance with between-group mean adjustment, reporting *mean±SD* per dimension.

**Results (pooled audience+experts).** As shown in Table 3, two effects emerge: (*i*) Introducing *hierarchy* (*setting1→setting2*) preferentially improves structure-centered dimensions, with pooled mean deltas +NS/+0.13, +RF/+0.16 (AE/+0.12, EE/+0.10), and a small trade-off on AT/−0.05. (*ii*) Adding *context engineering + bounded cycles* (*setting2→setting3*) strengthens expressivity and engagement, with deltas AE/+0.27, EE/+0.42, AT/+0.16, OE/+0.17 (NS/+0.06, RF/−0.03). Overall (*setting3* vs. *setting1*), pooled gains are AE/+0.40, EE/+0.52, NS/+0.20, AT/+0.10, OE/+0.25, RF/+0.12. These patterns reconcile both cohorts: the hierarchical step lifts NS/RF (audience signal), while context engineering closes the gap and yields the highest pooled AT/AE/EE/OE (expert signal corroborates). Within-prompt omnibus tests and baseline–vs–ours comparisons reported elsewhere point in the same directions.

**Takeaways.** *Hierarchy* accounts for the bulk of the improvement on narrative structure and pacing (NS/RF), while *context engineering + bounded cycles* unlocks camera/AV expressivity and subjective engagement (AT/AE/EE), culminating in the highest pooled *overall experience* (OE) in the full

Table 3: Ablation on pooled human ratings (audience + experts; subject-averaged across prompts P1–P3). Values are mean±SD; bold indicates the best per dimension.

| Model | NS | AT | AE | RF | EE | OE |
|---|---|---|---|---|---|---|
| setting1 flat | 2.90±0.48 | 3.01±0.36 | 2.63±0.47 | 2.96±0.36 | 2.67±0.56 | 2.77±0.50 |
| setting2 hier no ctx | **3.03**±0.54 | 2.96±0.51 | 2.76±0.64 | **3.11**±0.47 | 2.77±0.57 | 2.85±0.63 |
| setting3 full | 3.09±0.66 | **3.11**±0.48 | **3.03**±0.55 | 3.08±0.37 | **3.19**±0.56 | **3.02**±0.65 |

Pooled across cohorts with group-size weighting ($n_{aud}$=12, $n_{exp}$=4) and pooled variance including between-group mean adjustment. Rubric and prompts per Sec. §2.5; cohort protocols per Sec. 4.1–4.2.

framework. These ablations support that orchestration—rather than back-end choice—drives the observed quality gains under multi-prompt evaluation.

## 5 RELATED WORK

**Multi-agent orchestration and graph-level reflection.** LLM multi-agent frameworks organize role-specialized agents via scripted dialogue and tool use. Representative systems include CAMEL for communicative "societies" of agents (Li et al., 2023), AutoGen for multi-agent conversation and tool calling (Wu et al., 2024a), and MetaGPT for meta-programmed collaboration (Hong et al., 2024). Surveys catalog coordination topologies—central hubs, layered hierarchies, peer-to-peer meshes, and blackboards (Guo et al., 2024). Classic blackboard systems such as HEARSAY II established shared memory for opportunistic control (Erman et al., 1980; Nii, 1986). At scale, Mac-Net coordinates thousands of agents with DAGs and topological schedules (Qian et al., 2024), while ARG Designer begins to synthesize team topologies automatically (Li et al., 2025). Yet reflection largely remains a single-agent loop—e.g., Reflection and Self-Refine—rather than a property of the collaboration graph (Shinn et al., 2023; Madaan et al., 2023). The prevailing DAG assumption further inhibits downstream → upstream revision in creative pipelines. We introduce graph-level reflection with bounded cycles: directed graphs endowed with an explicit loop budget (a retry limit). Agents decide when and whom to re-message; once an edge's budget is exhausted, that edge is severed and execution reverts to a DAG. Retries reuse the same agent instance to preserve failure memory. A zero-loop budget recovers standard DAG execution.

**Organizational structure: centralization and hierarchy.** Although communication patterns are widely discussed, centralization and hierarchy are rarely treated as controllable variables in LLM MAS for media creation. Network science provides quantitative indices. Freeman centralization formalizes centralized versus decentralized organization (Freeman, 1978). Hierarchy can be assessed via Global Reaching Centrality (Mones et al., 2012) and graph theoretic hierarchy measures (Krackhardt, 2014). Empirical MAS studies often report topology choices qualitatively, such as hub and spoke, trees, and DAGs, without isolating how centralization and hierarchy levels impact throughput, robustness, and quality. We treat centralization and hierarchy as tunable hyperparameters of the agent graph, sweeping Freeman centralization and hierarchy indices such as GRC and Krackhardt's measure to quantify their effects on efficiency and failure containment. Combined with bounded cycles, this yields a factorial design that separates who coordinates from how feedback flows.

**Cross modal controllers and agentic video pipelines.** Language-as-controller systems connect LLM planners to specialist perception and generation tools across modalities. Typical examples include HuggingGPT for solving AI tasks with ChatGPT and its Hugging Face partners (Shen et al., 2023), MM ReAct for prompting ChatGPT for multimodal reasoning and action (Yang et al., 2023), Socratic Models for composing zero-shot multimodal reasoning with language (Zeng et al., 2022), and NExT-GPT as an any-to-any multimodal LLM (Wu et al., 2024b).

Film-oriented MAS include MovieAgent for automated movie generation via multi-agent CoT planning (Wu et al., 2025), FilmAgent for end-to-end film automation in virtual 3D spaces (Xu et al., 2025), Kubrick for multimodal agent collaborations for synthetic video generation (He et al., 2024), StoryAgent for customized storytelling video generation via multi-agent collaboration (Hu et al., 2024), LVAS Agent for long video–audio synthesis with multi-agent collaboration (Zhang et al., 2025b), and FilMaster bridges cinematic principles and generative AI with RAG driven camera language and audience centric postproduction, exporting OTIO timelines and introducing

*FilmEval* (Huang et al., 2025). These systems simulate studio roles yet typically hard-code pipelines or tree orchestrations and omit quantitative analysis of centralization, hierarchy, or graph-level reflection. At the planning bridge, storyboard and story visualization—exemplified by StoryGAN for sequential story visualization (Li et al., 2019)—inform text → shot design but stop short of end-to-end publish pipelines. Modern text-to-video backends, such as Lumiere for space–time diffusion video generation (Bar-Tal et al., 2024), Google Veo, and OpenAI Sora (Brooks et al., 2024), provide practical renderers callable by agents.

Nevertheless, the orchestration layer that bridges planning and rendering—how agents coordinate, where control is centralized, and whether graph-level feedback is permitted—remains underspecified. In this work, we systematically ablate the orchestration without claiming a full factorial sweep: we compare three internal versions (flat, hierarchical w/o context engineering, full) under identical back ends and prompts, and benchmark them against two black-box commercial systems using a standardized human rubric across three prompts. A broader factorial study over cycles and centralization/hierarchy is left to future work.

## 6 CONCLUSION

We presented **OmniAgent**, a cross–modal multi–agent framework for long–video generation that elevates the orchestration layer rather than any single backbone: (i) *hypergraph–based context collaboration* supplies on–demand shared context without inflating per–agent memory, (ii) *bounded cyclic execution* enables graph–level reflection under a retry budget, and (iii) *centralization and hierarchy* is quantified as controllable topology properties—bridging planning and rendering while mirroring film pipelines.

**Empirical findings.** Across three one–sentence prompts (1/2/3) and five versions (ours×3, commercial×2), audience ($n=12$) and experts ($n=4$) show consistent orchestration gains. *Audience:* pooled over prompts, *setting3_full* has the highest means on AT/AE/EE/OE, while *setting2_hier_no_ctx* leads NS/RF; Prompt 3 shows a significant model effect on AT; baseline–vs–ours favors our variants on RF/EE/OE in Prompt 2 and on NS/AE/OE in Prompt 3, with a reversal on RF in Prompt 1.[4] *Experts:* effects are modest yet align with audiences; in Prompt 3, model effects are significant for EE ($\chi^2=13.723$, $p=0.008$), NS ($\chi^2=12.121$, $p=0.016$), and OE ($\chi^2=11.015$, $p=0.026$), with AE trending ($p=0.052$); pooled across 1–3 (blocking expert×prompt), AE remains significant ($\chi^2=9.622$, $p=0.047$). Medians favor *setting3_full* on NS/AE, with *setting2_hier_no_ctx* comparable on RF (pairwise tests underpowered at $n=4$).

**Takeaways.** **(i)** Hierarchical orchestration improves structure–centric dimensions (NS/RF); **(ii)** adding hypergraph context and bounded cycles further boosts AE/EE/OE—closest to cinematic language and engagement; **(iii)** effectiveness is prompt–dependent, motivating multi–prompt evaluation.

**Limitations and future work.** Our human studies are modest in scale ($n=16$) and use three short prompts, which limits broad generalization across genres/lengths. Future work will expand prompts and rater diversity, ablate context depth and retry budgets, learn topologies end-to-end (e.g., autoregressive graph design), and integrate human-in-the-loop editing—bringing orchestration metrics (centralization, hierarchy) into adaptive, real-time controllers.

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

## A APPENDIX

### A.1 FILMEVAL EVALUATION INSTRUCTIONS

To enable fair and reproducible comparison with prior film–generation systems, we adopt the *FilmEval* instrument introduced by *FilMaster* Huang et al. (2025). This rubric organizes cinematic quality into six dimensions—Narrative & Script (NS), Audiovisuals & Techniques (AT), Aesthetics & Expression (AE), Rhythm & Flow (RF), Emotional & Engagement (EE), and Overall Experience

(OE)—decomposed into twelve criteria. We use the original questionnaire wording and five-point anchors for both *audience* and *expert* evaluations. For reporting, we also compute the rubric's two derived metrics—Camera Language (CL) and Cinematic Rhythm (CRh)—as deterministic combinations of the base criteria. Table 4 reproduces the rubric in a single consolidated table, and we further report inter-rater reliability (Cohen's $\kappa$) between audience and expert ratings.

Table 4: *FilmEval* rubric Unified presentation of all six dimensions (NS/AT/AE/RF/EE/OE) and twelve criteria with theexact Likert–5 anchors used in FILMASTER. We apply the same rubric to both *audience* and *expert* evaluations. Derived metrics—*Camera Language (CL)* and *Cinematic Rhythm (CRh)*—are computed from these base criteria exactly as specified in *FilmEval* Huang et al. (2025).

| FilmEval — Unified criteria and Likert-5 anchors for automatic evaluation and user study | |
|---|---|
| **Narrative & Script (NS)** | |
| **Script Faithfulness (SF)** | 1 point: Severely deviates from the original script; scenes and character settings completely inconsistent.
2 points: Partially follows the original script, with obvious deviations; multiple key settings changed.
3 points: Generally follows the original script, preserving main scenes and settings, but with details omitted.
4 points: Highly faithful, accurately reproduces most scenes and settings with rich details.
5 points: Completely faithful, precisely presents all scenes, settings, and details. |
| **Narrative Coherence (NC)** | 1 point: Chaotic story with serious logical contradictions and plot discontinuities.
2 points: Basically understandable but with multiple obvious logical gaps and coherence issues.
3 points: Generally coherent with minor deficiencies that do not affect main-plot understanding.
4 points: Smooth, coherent development with almost no obvious logical issues.
5 points: Completely coherent with clear cause–effect relationships and no logical holes. |
| **Audiovisuals & Techniques (AT)** | |
| **Visual Quality (VQ)** | 1 point: Severely broken visuals with numerous missing or distorted elements.
2 points: Obvious visual flaws; some scenes show missing or distorted elements.
3 points: Basically complete visuals with occasional minor errors not affecting viewing.
4 points: Clear and complete visuals with very few minor imperfections.
5 points: Flawless visuals; all elements perfectly rendered, no breakdowns. |
| **Character Consistency (CC)** | 1 point: Severely inconsistent character designs with dramatic appearance changes across scenes.
2 points: Noticeable fluctuations; features change in some scenes.
3 points: Generally consistent; occasional minor, unobtrusive inconsistencies.
4 points: Highly consistent across scenes and angles.
5 points: Perfectly consistent in all scenes and actions. |
| **Physical Law Compliance (PLC)** | 1 point: Severely violates physical laws; extremely unnatural movements/collisions/effects.
2 points: Multiple violations with obviously unrealistic movements/effects.
3 points: Generally compliant; a few motions/effects slightly artificial but acceptable.
4 points: Good compliance; natural movements; believable effects.
5 points: Perfect compliance; all motions/collisions/effects are highly realistic. |
| **Voice/Audio Quality (V/AQ)** | 1 point: Extremely poor audio; unclear VO and chaotic or missing SFX.
2 points: Poor audio; partially unclear VO and simple/inadequate SFX.
3 points: Average audio; basically clear VO and appropriate but unremarkable SFX.
4 points: Good audio; clear VO with rich SFX matching scenes.
5 points: Excellent audio; very clear, vivid VO and rich, nuanced SFX with great expressiveness. |
| **Aesthetics & Expression (AE)** | |
| **Cinematic Techniques (CT)** | 1 point: Single, stiff shots; no variation; lacks basic film language.
2 points: Limited shot variation; stiff camera; poor film language.
3 points: Common techniques; basically smooth camera; basic expression.
4 points: Rich film language; smooth/natural camera; reasonable and effective variations.
5 points: Highly creative shot usage; precise camera; rich variations with exceptional expressiveness. |
| **Audio–Visual Richness (AVR)** | 1 point: Extremely limited expression; monotonous/repetitive A/V elements; minimal variation/layering.
2 points: Some attempts but overall formulaic; little dynamic or stylistic variation.
3 points: Moderate diversity; richness uneven and lacks coherence or artistic depth.
4 points: Visually and sonically expressive; multiple techniques used effectively for layered meaning/mood.
5 points: Exceptionally rich; diverse, inventive, highly expressive A/V language with strong impact. |
| **Rhythm & Flow (RF)** | |
| **Narrative Pacing (NP)** | 1 point: Completely uncontrolled; too fast or too slow; severely hurts comprehension.
2 points: Obviously inconsistent; some developments too quick or too slow.
3 points: Generally appropriate pacing with reasonable progression.
4 points: Well-controlled pacing; natural progression; good tension–relief balance.
5 points: Precisely controlled pacing serving the story and capturing audience emotions. |
| **Video–Audio Coordination (VAC)** | 1 point: Severely unsynchronized A/V; completely mismatched lip-sync.
2 points: Clear lack of synchronization; poor coordination between voice and visuals.
3 points: Basically synchronized with occasional inconsistencies.
4 points: Good coordination; sound matches visual actions well.
5 points: Perfect synchronization; all sound elements precisely match visual actions. |
| **Emotional & Engagement (EE)** | |
| **Compelling Degree (CD)** | 1 point: No appeal; difficult to feel immersed or emotionally connected.
2 points: Insufficient appeal; weak emotional rendering; hard to maintain attention.
3 points: Basic appeal that can raise interest but lacks deep resonance.
4 points: Strong appeal with effective emotional rendering that elicits resonance.
5 points: Extremely compelling with powerful tension and sustained engagement. |
| **Overall Experience (OE)** | |
| **Overall Quality (OQ)** | 1 point: Extremely poor across multiple dimensions; lacks viewing value.
2 points: Poor with key dimensions performing badly; limited viewing value.
3 points: Average performance across dimensions; basic viewing value.
4 points: Good performance with dimensions working well together; high viewing value.
5 points: Outstanding across all dimensions with excellent coordination; extremely high artistic/viewing value. |

## A.2 WITHIN-PROMPT FRIEDMAN

Table 5: Within-prompt Friedman tests (model main effect) per dimension. df= 4 for all tests.

| Prompt | Dimension | $\chi^2$ | $p$ | $W$ | $N$ |
|--------|-----------|----------|------|------|-----|
| P1 | NS | 1.83 | 0.767 | 0.038 | 12 |
| P1 | AT | 3.64 | 0.457 | 0.076 | 12 |
| P1 | AE | 1.64 | 0.802 | 0.034 | 12 |
| P1 | RF | 8.30 | 0.081$^\dagger$ | 0.173 | 12 |
| P1 | EE | 2.79 | 0.593 | 0.058 | 12 |
| P1 | OE | 0.88 | 0.927 | 0.018 | 12 |
| P2 | NS | 3.25 | 0.517 | 0.074 | 12 |
| P2 | AT | 2.51 | 0.644 | 0.057 | 12 |
| P2 | AE | 4.07 | 0.397 | 0.092 | 12 |
| P2 | RF | 5.92 | 0.206 | 0.134 | 12 |
| P2 | EE | 6.83 | 0.145 | 0.155 | 12 |
| P2 | OE | 3.35 | 0.501 | 0.076 | 12 |
| P3 | NS | 7.65 | 0.105 | 0.159 | 12 |
| P3 | AT | 9.60 | 0.048* | 0.200 | 12 |
| P3 | AE | 9.01 | 0.061$^\dagger$ | 0.188 | 12 |
| P3 | RF | 7.99 | 0.092 | 0.167 | 12 |
| P3 | EE | 8.07 | 0.089$^\dagger$ | 0.168 | 12 |
| P3 | OE | 6.95 | 0.138 | 0.145 | 12 |

$^*p<.05$, $^\dagger p<.10$ (two-sided; Kendall's $W$ reported for concordance).

## A.3 BASELINES VS OURS

Table 6: Commercial baselines (mean of *aipai*, *video_ocean*) vs. pooled Ours (mean of *setting1/2/3*) by prompt. Entries are $\Delta M$=Ours − Baselines with paired Wilcoxon $p$.

| Prompt | Dimension | $\Delta M$ | $p$ | Mark |
|---|---|---|---|---|
| P1 | NS | −0.10 | 0.470 | |
| P1 | AT | −0.16 | 0.366 | |
| P1 | AE | −0.10 | 0.733 | |
| P1 | RF | −0.42 | 0.030* | Baselines > Ours |
| P1 | EE | 0.06 | 0.752 | |
| P1 | OE | −0.07 | 0.844 | |
| P2 | NS | 0.31 | 0.135 | |
| P2 | AT | 0.31 | 0.146 | |
| P2 | AE | 0.54 | 0.064$^\dagger$ | |
| P2 | RF | 0.49 | 0.028* | Ours > Baselines |
| P2 | EE | 0.60 | 0.012* | Ours > Baselines |
| P2 | OE | 0.56 | 0.037* | Ours > Baselines |
| P3 | NS | 0.78 | 0.034* | Ours > Baselines |
| P3 | AT | 0.68 | 0.052$^\dagger$ | |
| P3 | AE | 0.65 | 0.034* | Ours > Baselines |
| P3 | RF | 0.74 | 0.071$^\dagger$ | |
| P3 | EE | 0.65 | 0.064$^\dagger$ | |
| P3 | OE | 0.78 | 0.045* | Ours > Baselines |

$^*p<.05$, $^\dagger p<.10$ (two-sided Wilcoxon; paired within-subject).

## A.4 WITHIN-PROMPT FRIEDMAN TESTS

Table 7: Experts: within-prompt Friedman tests per dimension (5 models). We report $\chi^2$, $df=4$, $p$, and Kendall's W ($\eta_c$); $N=4$ experts per prompt.

| Prompt | Dimension | $\chi^2$ | df | p | $\eta_c$ |
|---|---|---|---|---|---|
| | NS | 9.389 | 4 | 0.052$^\dagger$ | 0.587 |
| | AT | 4.213 | 4 | 0.378 | 0.263 |
| Prompt 1 | AE | 2.648 | 4 | 0.618 | 0.165 |
| | RF | 4.271 | 4 | 0.371 | 0.267 |
| | EE | 1.477 | 4 | 0.831 | 0.092 |
| | OE | 2.098 | 4 | 0.718 | 0.131 |
| | NS | 7.014 | 4 | 0.135 | 0.438 |
| | AT | 3.429 | 4 | 0.489 | 0.214 |
| Prompt 2 | AE | 6.154 | 4 | 0.188 | 0.385 |
| | RF | 3.507 | 4 | 0.477 | 0.219 |
| | EE | 5.784 | 4 | 0.216 | 0.361 |
| | OE | 7.833 | 4 | 0.098$^\dagger$ | 0.490 |
| | NS | 12.121 | 4 | 0.016$^*$ | 0.758 |
| | AT | 7.514 | 4 | 0.111 | 0.470 |
| Prompt 3 | AE | 9.412 | 4 | 0.052$^\dagger$ | 0.588 |
| | RF | 7.032 | 4 | 0.134 | 0.440 |
| | EE | 13.723 | 4 | 0.008$^*$ | 0.858 |
| | OE | 11.015 | 4 | 0.026$^*$ | 0.688 |

$^*p<.05$, $^\dagger p<.10$. With $N=4$ raters, high Kendall's W values (e.g., $>0.6$) indicate strong within-panel agreement.

## A.5 BASELINES VS. OURS

Table 8: Experts: pooled commercial baselines (*aipai*, *video_ocean*) vs. pooled ours (*setting1/2/3*). Entries are $\Delta M$=Ours−Baseline with paired Wilcoxon $p$.

| Prompt | Dim | Ours Mean | Base Mean | $\Delta M$ | $p$ | Note |
|--------|-----|-----------|-----------|------------|-----|------|
|   | NS | 2.792 | 3.250 | −0.458 | 0.125 |   |
|   | AT | 3.250 | 3.281 | −0.031 | 1.000 |   |
| A | AE | 3.000 | 3.125 | −0.125 | 0.625 |   |
|   | RF | 3.333 | 3.563 | −0.229 | 0.375 |   |
|   | EE | 3.000 | 3.250 | −0.250 | 0.875 |   |
|   | OE | 3.167 | 3.375 | −0.208 | 0.625 |   |
|   | NS | 2.917 | 3.563 | −0.646 | 0.285 |   |
|   | AT | 3.104 | 3.281 | −0.177 | 0.875 |   |
| B | AE | 2.708 | 3.125 | −0.417 | 0.285 |   |
|   | RF | 2.917 | 3.250 | −0.333 | 0.625 |   |
|   | EE | 2.917 | 3.250 | −0.333 | 0.625 |   |
|   | OE | 2.833 | 3.250 | −0.417 | 0.414 |   |
|   | NS | 3.542 | 2.125 | 1.417 | 0.125 |   |
|   | AT | 3.125 | 2.344 | 0.781 | 0.125 |   |
| C | AE | 3.208 | 2.250 | 0.958 | 0.125 |   |
|   | RF | 3.458 | 2.688 | 0.771 | 0.125 |   |
|   | EE | 3.500 | 2.000 | 1.500 | 0.125 |   |
|   | OE | 3.417 | 2.125 | 1.292 | 0.125 |   |

## B TEST PROMPTS

To evaluate the capability of our multi-agent framework in generating coherent, stylistically diverse long-form videos, we designed three textual prompts. Each prompt is approximately 36 words long and specifies a target duration of about one minute, along with distinct visual and narrative requirements. The prompts cover different genres and modalities—live-action, 2D animation, and fantasy action—to assess the system's adaptability across styles.

**Prompt 1**

1 minute realistic movie scene: The agent sneaked into the heavily guarded laboratory at night, avoiding the patrolling guards, and quietly approached the safe where the confidential documents were stored.

**Prompt 2**

1 minute 2D animation clip: A young boy and a young girl are walking and playing around on a flower-filled riverbank, expressing their feelings for each other.

**Prompt 3**

1 minute short video: The human warrior holds a shield to block the orc charge in the canyon. The two sides fight until they reach a broken bridge. The warrior swings his sword to knock down the orc's weapon, and the orc counterattacks, forcing the warrior to fall off the cliff.

## C USE OF LARGE LANGUAGE MODELS (LLMS)

To comply with ICLR's guidance on the use of LLMs, we disclose that a large language model (LLM)–based writing assistant was used *only* for grammar and style editing of this manuscript. The scope and limits are as follows.

**Scope of assistance.** The LLM was used to (i) fix grammar, spelling, and minor punctuation; (ii) improve sentence clarity and readability; (iii) harmonize terminology (e.g., consistently using "Prompts 1/2/3" and model names across sections); and (iv) assist with minor LaTeX formatting

(line breaks, table captions, and cross-reference wording). All suggestions were reviewed and edited by the authors.

**What the LLM *was not* used for.** The LLM did *not* contribute to research conception, experiment or system design, data collection, analysis, or result interpretation; it did not generate figures, tables, numerical results, literature content, or citations. No prompts, code, datasets, or evaluation artifacts were produced or modified by the LLM.

**Authorship, responsibility, and verification.** The authors take full responsibility for all content written in their names, including any text that was edited following LLM suggestions. All technical statements, references, equations, and claims were verified by the authors. The LLM is not an author and does not qualify for authorship.

**Privacy and double-blind considerations.** To preserve double-blind review, we do not disclose the specific vendor or model. Only manuscript text was provided to the assistant; no private datasets, identities, reviews, or sensitive information were shared.

**Reproducibility.** The disclosed LLM usage does not affect the reproducibility of our methods or results. All experiments, prompts, models, and evaluation protocols are fully specified in the main text and appendix.

