# OpenReview forum: "Omniagent: Long-Video Generation via Cross-Modal Multi-Agent Orchestration"
_ICLR.cc/2026/Conference — ICLR 2026 Conference Withdrawn Submission_

### Official Review · Reviewer_QpHb · 2025-10-25

**Soundness:** 3
**Presentation:** 2
**Contribution:** 2
**Rating:** 6
**Confidence:** 4

**Summary:**

This paper proposes OmniAgent, a hierarchical, graph-based multi-agent framework for long video generation. The hypergraph-based context retrieval mechanism is introduced that enables on-demand, collaborative knowledge gathering across agents, balancing context richness with memory efficiency. A controlled cyclic execution strategy is proposed, which allows limited backward edges for iterative refinement and reflection.

**Strengths:**

1. The idea of integrating DAG and context engineering into multi-agent systems is interesting and inspiring.
2. The evaluation is extensiv and thorough.

**Weaknesses:**

1. The differences among multi-agent work for video generation (e.g. MovieAgent) are not clear.

2. The problems aimed to be solved are not clear in the paper.

3. Rely on advanced models (e.g. GPT-4o). What if these advanced models are not available or evolve? What are the performances with less advanced models?

4. In my view, I do not see much difference in qualitative videos compared to baselines (in supplementary). As the opinion is subjective, I request that the qualitative results of the comparison be properly explained.

**Questions:**

1. Does the paper aim to mimic the film production workflow in real world, or aim to solve the problems of current multi-agent works? If is to mimic the workflow, why are the designs of hyper graph-based context collaboration matter? Other designs (hierarchical CoT in MovieAgent) may take effect.

2. What are the effects of private memory?

---

### Official Review · Reviewer_CwEm · 2025-10-29

**Soundness:** 3
**Presentation:** 3
**Contribution:** 2
**Rating:** 2
**Confidence:** 4

**Summary:**

The paper proposes OmniAgent, a hierarchical, graph-basedmulti-agent framework for long video generation. It integrates LLM-driven agents with multimodal generation tools to enable modular specialization and scalable inter-agent collaboration. It also proposes hypergraph nodes to enable temporary group discussions among agents lacking sufficient context. Besides, it allows iterative refinement by directed cyclic graphs with limited retries.
Experiments show that the full OmniAgent system achieves better performance.

**Strengths:**

1.	The paper clearly explains its design choices, and the analogy to real-world film production is intuitive and well presented.

2.	The paper proposes a new graph-based framework to improve multi-agent collaboration.

3.	Human evaluation is well-structured, incorporating both expert and audience raters to ensure comprehensive assessment.

**Weaknesses:**

1.	The proposed framework is mainly a combination of existing ideas, lacking novelty. The contribution is therefore more of a system integration than a new algorithmic or theoretical advance.

2.	The experiments only compare against commercial video-generation APIs, not against academic multi-agent frameworks, e.g., MovieAgent: Automated Movie Generation via Multi-Agent CoT Planning; Mora: Enabling Generalist Video Generation via A Multi-Agent Framework; AniMaker: Multi-Agent Animated Storytelling with MCTS-Driven Clip Generation; etc.

3.	The paper lacks clarity on how hypergraph “team meetings” are implemented, how consensus is formed, and how reflection loops affect latency and resource cost.

**Questions:**

1.	How is the retry budget (Rmax=3) chosen? Did the authors try other values?

2.	Can OmniAgent outperform other research frameworks (e.g., MovieAgent, Mora, AniMaker)?

3.	What is the computational cost of the proposed pipeline?

---

### Official Review · Reviewer_2XBY · 2025-10-31

**Soundness:** 2
**Presentation:** 2
**Contribution:** 3
**Rating:** 2
**Confidence:** 4

**Summary:**

This paper presents a multi-agent framework, OmniAgent, for long video generation. This framework is motivated by real-world creative workflows, introducing hypergraph-based context collaboration and bounded loop feedback for efficient context retrieval and iterative refinement. Experiments evaluate OmniAgent's performance on minute-level video generation from single-sentence prompts. Both the ablation analysis and the comparison with commercial models demonstrate the effectiveness of the proposed framework.

**Strengths:**

1. The graph-based formulation of multi-agent systems established in this work is clear and meaningful, providing a basic framework for subsequent hypergraph cooperation and cyclic refinement.

2. The proposed hierarchical, graph-based multi-agent framework is effective, facilitating a modular and specialized pipeline for long video production.

**Weaknesses:**

1. The experimental evaluation in this paper is quite inadequate. Only three text prompts are used for evaluation, which are likely cherry-picked and do not fully encompass different contents and styles. Besides, the paper provides no qualitative comparisons, which is unusual for a video generation work (although some videos are provided in the supplementary materials). Analysis beyond the ablation study is also missing, such as the impact of hyperparameters maximum depth and retry budget on performance.

2. The paper provides no illustrations. This hinders the reader's understanding of the methodology, especially given the complexity of a multi-agent framework with hypergraph-based connections and cycles.

3. In particular, for Section 2.2, the overly brief description and lack of illustrations make it difficult to understand the detailed implementation of the proposed hierarchical workflow graph.

4. Technically, controlled cyclic execution is not a particularly novel design. Existing works (such as GenMAC) have also introduced similar iterative refinement mechanisms.

**Questions:**

It is recommended that the authors carefully polish the paper, providing necessary illustrations and supplementing any missing experimental results.

---

### Note · Authors · 2025-11-12

I have read and agree with the venue's withdrawal policy on behalf of myself and my co-authors.